# *RAD51D* Aberrant Splicing in Breast Cancer: Identification of Splicing Regulatory Elements and Minigene-Based Evaluation of 53 DNA Variants

**DOI:** 10.3390/cancers13112845

**Published:** 2021-06-07

**Authors:** Elena Bueno-Martínez, Lara Sanoguera-Miralles, Alberto Valenzuela-Palomo, Víctor Lorca, Alicia Gómez-Sanz, Sara Carvalho, Jamie Allen, Mar Infante, Pedro Pérez-Segura, Conxi Lázaro, Douglas F. Easton, Peter Devilee, Maaike P. G. Vreeswijk, Miguel de la Hoya, Eladio A. Velasco

**Affiliations:** 1Splicing and Genetic Susceptibility to Cancer Laboratory, Unidad de Excelencia Instituto de Biología y Genética Molecular, Consejo Superior de Investigaciones Científicas (CSIC-UVa), 47003 Valladolid, Spain; elena.bueno@uva.es (E.B.-M.); lara.sanoguera@uva.es (L.S.-M.); alberto.valenzuela@ibgm.uva.es (A.V.-P.); 2Molecular Oncology Laboratory CIBERONC, IdISSC (Instituto de Investigación Sanitaria del Hospital Clínico San Carlos), Hospital Clinico San Carlos, 28040 Madrid, Spain; victor.lorca@salud.madrid.org (V.L.); agomezsanz@salud.madrid.org (A.G.-S.); pedro.perez@salud.madrid.org (P.P.-S.); 3Centre for Cancer Genetic Epidemiology, Department of Public Health and Primary Care, University of Cambridge, Cambridge CB1 8RN, UK; sc2017@medschl.cam.ac.uk (S.C.); jma73@medschl.cam.ac.uk (J.A.); dfe20@medschl.cam.ac.uk (D.F.E.); 4Cancer Genetics, Unidad de Excelencia Instituto de Biología y Genética Molecular (CSIC-UVa), 47003 Valladolid, Spain; minfante@ibgm.uva.es; 5Hereditary Cancer Program, Catalan Institute of Oncology, IDIBELL and CIBERONC, 08908 Hospitalet de Llobregat, Spain; clazaro@iconcologia.net; 6Department of Human Genetics, Leiden University Medical Center, 2300RC Leiden, The Netherlands; P.Devilee@lumc.nl (P.D.); M.P.G.Vreeswijk@lumc.nl (M.P.G.V.)

**Keywords:** breast cancer, ovarian cancer, susceptibility genes, RAD51D, ESE, ESS, aberrant splicing, VUS, minigene, clinical interpretation

## Abstract

**Simple Summary:**

In the BRIDGES project, the breast/ovarian cancer gene *RAD51D* has been sequenced in >113,000 women. In the present study, we focused on the impact that 11 pre-selected *RAD51D* variants at the intron/exon boundaries had on the splicing process (intron removal). For this purpose, we developed a splicing reporter minigene, containing *RAD51D*-exons 2–9 wherein any variant could be introduced and functionally assayed for splicing alterations. All variants impaired splicing, 10 of which caused complete splicing aberrations. Moreover, we developed a minigene-based strategy to search for non-canonical, spliceogenic variants that disrupted splicing enhancers/silencers in the non-constitutive exon 3. Twenty-six BRIDGES and 16 artificial exon 3 variants were also tested. Thirty variants impaired splicing by producing variable amounts of the FL transcript. In total, up to 9 variants were classified as Likely Pathogenic, and therefore were clinically actionable. Carriers may benefit from tailored prevention protocols and therapies.

**Abstract:**

*RAD51D* loss-of-function variants increase lifetime risk of breast and ovarian cancer. Splicing disruption is a frequent pathogenic mechanism associated with variants in susceptibility genes. Herein, we have assessed the splicing and clinical impact of splice-site and exonic splicing enhancer (ESE) variants identified through the study of ~113,000 women of the BRIDGES cohort. A RAD51D minigene with exons 2–9 was constructed in splicing vector pSAD. Eleven BRIDGES splice-site variants (selected by MaxEntScan) were introduced into the minigene by site-directed mutagenesis and tested in MCF-7 cells. The 11 variants disrupted splicing, collectively generating 25 different aberrant transcripts. All variants but one produced negligible levels (<3.4%) of the full-length (FL) transcript. In addition, ESE elements of the alternative exon 3 were mapped by testing four overlapping exonic microdeletions (≥30-bp), revealing an ESE-rich interval (c.202_235del) with critical sequences for exon 3 recognition that might have been affected by germline variants. Next, 26 BRIDGES variants and 16 artificial exon 3 single-nucleotide substitutions were also assayed. Thirty variants impaired splicing with variable amounts (0–65.1%) of the FL transcript, although only c.202G>A demonstrated a complete aberrant splicing pattern without the FL transcript. On the other hand, c.214T>C increased efficiency of exon 3 recognition, so only the FL transcript was detected (100%). In conclusion, 41 *RAD51D* spliceogenic variants (28 of which were from the BRIDGES cohort) were identified by minigene assays. We show that minigene-based mapping of ESEs is a powerful approach for identifying ESE hotspots and ESE-disrupting variants. Finally, we have classified nine variants as likely pathogenic according to ACMG/AMP-based guidelines, highlighting the complex relationship between splicing alterations and variant interpretation.

## 1. Introduction

*RAD51D* [MIM#602954] is one of the five RAD51 paralogs (*RAD51B, RAD51C, RAD51D*, *XRCC2*, *XRCC3*) that play an important role in the repair of DNA double-strand breaks via homologous recombination [1,2]. *RAD51D* loss-of-function variants confer risk of breast and/or ovarian cancer [3,4]. In a cohort of 6690 families, *RAD51D* pathogenic variants were associated with relative risks of 7.6 and 1.83 for tubo-ovarian cancer and breast cancer, respectively [3]. Recently, two large-scale studies have estimated an overall breast cancer relative risk of 1.8 and 1.72, respectively [5,6]. The association with triple-negative (TN) breast cancer (RR = 6.1) appears to be particularly strong [5].

During RNA splicing, introns from eukaryotic genes are removed from pre-mRNA, and consecutive exons are precisely joined together to form mature mRNA [7]. Exon recognition is a crucial step controlled by a large number of trans-acting factors, including ribonucleoproteins and splicing factors as well as cis-acting sequences identified by splicing machinery. The array of splicing signals is extensive and includes 5′ and 3′ splice sites, the polypyrimidine tract, the branch point and exonic and intronic splicing enhancers (ESE/ISE) and silencers (ESS/ISS) that stimulate or repress exon inclusion in mature mRNA [8]. It has been proposed that ~60% of disease-causing variants disrupt pre-mRNA processing [9], generating aberrant transcripts that can affect protein function and correlate with an increased risk of a given genetic disease. Consequently, it is imperative to assess the biological and clinical significance of spliceogenic variants in order to improve genetic diagnosis and counseling as well as to trigger new therapeutic interventions [10]. Unfortunately, the lack of accurate in silico predictors makes functional assays vital to doing so. Next, splicing functional assays, either from patient or minigene RNAs, provide critical data to evaluate the pathogenicity of a particular genetic variant [11,12,13,14,15].

Our study was conducted using the framework of the BRIDGES project (Breast Cancer After Diagnostic Gene Sequencing; https://bridges-research.eu/, accessed on 1 April 2021), which has sequenced a panel of 34 known or suspected breast cancer susceptibility genes in 60,466 cases and 53,461 controls [5]. We bioinformatically analyzed 47 *RAD51D* variants from the intron/exon boundaries, 11 of which were selected and functionally tested in a *RAD51D*-splicing reporter minigene including exons 2–9 (mgR51D_ex2-9). In addition, we functionally mapped ESE-rich regions of the non-constitutive exon 3 by overlapping microdeletions and analyzed 42 candidate BRIDGES and artificial ESE variants. Finally, we made a tentative clinical classification of BRIDGES variants according to ACMG/AMP (American College of Medical Genetics and Genomics and the Association for Molecular Pathology)-based guidelines.

## 2. Materials and Methods

Ethical approval for this study was obtained from the Ethics Committee of the Spanish National Research Council-CSIC (28 May 2018).

### 2.1. Variant and Transcript Annotations

Forty-seven variants from the intron–exon boundaries (3′ splice-site: intron/exon [IVS-10_IVS-1/2nt]; 5′ splice-site: exon/intron [2nt/IVS+1_IVS+10]) [16] and 34 from RAD51D exon 3 were collected from the BRIDGES sequencing data [5]. RNA outcomes and predicted protein products were described according to the Human Genome Variation Society guidelines (http://varnomen.hgvs.org/, accessed on 1 April 2021) using Ensembl reference transcript ID ENST00000345365.10 (GenBank NM_002878.3). To simplify, we also annotated splicing events, as previously described [17].

### 2.2. Bioinformatics

The workflow of the splicing assays is outlined in Appendix A. To identify potential splicing variants, in silico studies were performed using MaxEntScan (MES, http://hollywood.mit.edu/burgelab/maxent/Xmaxentscan_scoreseq.html, accessed on 1 April 2021) (cut-off ≥ 3.0) [18] and NNSplice (https://www.fruitfly.org/seq_tools/splice.html, accessed on 1 April 2021) [19] when MES was not informative. Potential spliceogenic variants were selected according to the following criteria: (i) MES score changes (>15%) [16,20] and (ii) creation of alternative sites (Appendix A).

Four in silico approaches were used to predict variant-induced modifications in splicing regulatory elements: (a) HEXplorer (ΔHZ_EI_; cut-off < −5) (https://www2.hhu.de/rna/html/hexplorer_score.php, accessed on 1 April 2021) [21]; (b) Hot-Skip (https://hot-skip.img.cas.cz/, cut-off ≥ 1, accessed on 1 April 2021) or Ex-Skip (https://ex-skip.img.cas.cz/, accessed on 1 April 2021) [22]; (c) calculation of total ESRseq score changes (ΔtESRseq) by quantitative assessment of RNA hexamers (cut-off < −0.75) [23]; and (d) alteration of the ESE/ESS balance by HSF at Genomnis (https://hsf.genomnis.com/home, accessed on 1 April 2021).

### 2.3. Minigene Construction and Site-Directed Mutagenesis

A 3591-bp fragment including *RAD51D* exons 2 to 9 and flanking introns was generated by DNA synthesis (Genewiz, South Plainfield, NJ, USA). This insert was cloned into the splicing plasmid pSAD [24,25,26] (minigene mgR51D_ex2-9, Figure 1, Appendix A). All sequence variations were introduced into mgR51D_ex2-9 by site-directed mutagenesis using the QuikChange Lightning kit (Agilent, Santa Clara, CA, USA) (Appendix A). All constructs were confirmed by Sanger sequencing (Macrogen, Madrid, Spain).

### 2.4. Functional Assays

MCF-7 cell growth, transfection and inhibition of the nonsense-mediated decay were performed as previously described [16]. The triple-negative breast cancer (TNBC) cell line MDA-MB-231 was cultured in Dulbecco’s Modified Eagle Medium (DMEM) supplemented with 10% fetal bovine serum, 2 mM glutamine, 1% non-essential amino acids and 1% penicillin/streptomycin solution.

RNA was purified using the Genematrix Universal RNA Purification Kit (EURx, Gdansk, Poland), with on-column DNAse I digestion. RNA (400 ng) was retrotranscribed using the vector exon V2-specific primer RTPSPL3-RV (5′-TGAGGAGTGAATTGGTCGAA-3′) and the RevertAid First-Strand cDNA Synthesis Kit (Life Technologies, Carlsbad, CA, USA), following the manufacturer’s instructions. Two μL of cDNA were used for amplification of the regions of interest using Platinum Taq polymerase (Life Technologies). To increase specificity, cDNA was amplified, with one primer located in one *RAD51C* exon of the insert and another one in a vector exon (V1 or V2). Thus, for variants of the donor site of exon 2, the amplification was performed using primers SD6-PSPL3_RTFW (5′-TCACCTGGACAACCTCAAAG-3′) and RTR51D_9-rv (5′-GTCCCTGTCTCGAGTTATG-3′) (amplicon size = 843 bp). For the remaining variants (exons 3 to 8), the following primers were used: RTR51D_ex2-fw (5′-TAGCTCAGAAATGTGGCTT-3′) and RTpSAD-RV (Patent P201231427, CSIC) (size = 885 bp). Samples were denatured at 94 °C for 2 min, followed by 35 cycles × (94 °C, 30 s/59 °C, 30 s/72 °C, 1 min/kb) and 72 °C for 5 min. RT-PCR products were sequenced by Macrogen (Madrid, Spain).

To estimate the relative proportions of each transcript, semi-quantitative fluorescent RT-PCRs were performed using a FAM-labeled primer under standard conditions, except that 26 cycles were herein applied [14,27,28]. Fluorescent products were run with LIZ-1200 Size Standard by Macrogen (Seoul, Korea) and analyzed using Peak Scanner software V1.0 (Life Technologies). The mean peak areas of three independent experiments of each variant were used to calculate the relative proportions of each transcript as well as the standard deviations.

### 2.5. ACMG/AMP-Like Classification of 37 RAD51D Variants Detected in BRIDGES Samples

No *RAD51D* expert panel specifications of the American College of Medical Genetics and Genomics or the Association for Molecular Pathology (ACMG/AMP) variant curation guidelines were currently available (www.clinicalgenome.org, last accessed on 29 December 2020). Therefore, we performed a tentative classification based on: (i) generic ACMG/AMP guidelines [29], (ii) specific recommendations of the ClinGen Sequence Variant Interpretation Working Group for interpreting the loss-of-function PVS1 [30] and functional PS3/BS3 evidence codes [31], (iii) some non-gene-specific approaches proposed by the ClinGen *CDH1* variant curation expert panel [32] and (iv) ad hoc rules based on expert judgment. The latter were essential to assign a specific functional splicing code (PS3/BS3) to complex minigene readouts (Appendix A). In addition to PS3/BS3, only the rarity code (PM2) made a significant contribution to the classification process.

## 3. Results

### 3.1. Bioinformatics Analysis and Functional Assays

Forty-seven *RAD51D* variants detected in BRIDGES subjects were collected from the intron–exon boundaries of exons 2 to 9 and analyzed using MaxEntScan (MES) or NNSplice (Appendix A). Eleven potential splice-site disrupting variants were selected to be assayed in MCF-7 cells (Table 1).

A RAD51D minigene with exons 2–9 was constructed in splicing vector pSAD (minigene mgR51D_ex2–9, Figure 1, Appendix A). Fragment analysis of the wild type (wt) minigene revealed the full-length (FL) transcript (V1-RAD51D exons 2 to 9-V2) and several alternative isoforms (Table 1; Figure 1C), of which ∆(E2_5), ∆(E3_5) and ∆(E4_5), among others, were formerly reported as naturally occurring transcripts [33]. All 11 variants impaired splicing, wherein the FL transcript was (almost) absent in 10 variants and c.263+6T>C still expressed a considerable level of FL transcript (49%). Eight of these variants (c.263+6T>C, c.343C>T, c.345+2T>C, c.480+1G>A, c.476_480+1dup, c.481-8C>A, c.577-2A>G and c.738+1G>A) were also tested in the triple-negative breast cancer cells MDA-MB-231, in which they replicated the splicing profiles (Appendix A).

Five variants disrupted the canonical 3′ splice site (3′SS), although c.83-4_83-3delinsAG and c.481-8C>A also created a de novo site (transcripts (E2p2) and ▼(E6p6), respectively), and variant c.577-2A>G mainly provoked the use of an intronic cryptic 3′SS (▼(E7p41)). Four variants disrupted the canonical 5′ splice site (5′SS): c.345+2T>C, c.480+1G>A, c.476_480+1dup and c.738+1G>A. Variants c.476_480+1dup and c.738+1G>A also caused the recognition of alternative 5’SS (▼(E5q6) and ▼(E8q43), respectively). Finally, c.263+6T>C and c.343C>T weakened the natural 5′SS of exons 3 and 4, respectively.

### 3.2. Transcript Analysis

Our minigene analysis identified 25 transcripts, including the alternative ones generated by the wt minigene (Appendix A and Appendix A). Exon (or multi-exon) skipping was a frequent outcome, with 13 different aberrant transcripts detected: ∆(E2), ∆(E2_3), ∆(E2_5), ∆(E3), ∆(E3_5), ∆(E3_7), ∆(E4), ∆(E4_5), ∆(E4_7), ∆(E5), ∆(E6_9), ∆(E7) and ∆(E8). Another five aberrant transcripts were due to the use of de novo splice sites (▼(E2p2), ▼(E5q6), ▼(E6p6), ▼(E7p41) and ▼(E8q43)), and five aberrant transcripts combined skipping and alternative site usage events ((∆(E2) ▼(E5q6)), (∆(E3) ▼(E5q6)), (∆(E3_5) ▼(E6p6)), (∆(E4) ▼(E5q6)) and (∆(E4_5) ▼(E6p6)). Two additional fragments of 487- and 1363-nt were also found, but the presumed aberrant events could not be characterized. Seventeen transcripts introduced PTCs that were predicted to inactivate RAD51D, and six kept the reading frame (Table 1, Appendix A).

### 3.3. ESE Mapping

Effective exon recognition in alternative splicing requires the binding of trans-acting factors to exon-splicing enhancer (ESE) sequences [34]. Exon 3 is a non-constitutive exon, excluded from several *RAD51D* naturally occurring, alternatively spliced isoforms [33] and consequently, it is likely regulated by splicing regulatory elements (SRE) [35]. We proceeded to functionally map SREs in exon 3 by introducing four overlapping 30–35-bp deletions (c.147_176del, c.172_206del, c.202_235del and c.231_260del) along this exon (119 bp), excluding the first two and last three nucleotides to preserve splice site-conserved positions [24,25]. All four deletions had an impact on splicing, and different levels of exon 3 skipping were observed (Figure 2A,B; Appendix A; Appendix A), pointing out that these sequences might have contained regulatory motifs. Deletion c.202_235del affected splicing the most, observing no trace of the FL transcript (Figure 2A,B).

We then decided to functionally assay 25 BRIDGES variants reported at the most spliceogenic segment, c.202_235del, and surrounding sequences. In this study, variants with a FL transcript proportion ranging from 65.8 to 73.1% (<10% reduction of the FL transcript in the wt minigene) were not considered spliceogenic (see Section 4). In total, 16 out of 25 variants (64%) effectively disrupted splicing (Table 2; Figure 2A,B; Appendix A), wherein exon 3 skipping (not observed in the wt minigene) was detected and ∆(E3_5) was increased (>17.5%) in all the spliceogenic variants. In addition, c.202G>A also expressed a new isoform, ∆(E3p36), through the use of an internal cryptic acceptor site. Remarkably, c.202G>A was the only variant that did not produce FL transcripts, whereas c.214T>C produced 100% full-length transcripts.

To refine the SRE map and identify potential hotspots for spliceogenic variants, five additional overlapping 10/11-bp deletions of the c.202_235 interval were tested. All deletions impaired splicing, being particularly significant in c.210_219del, wherein the FL transcript was absent (Appendix A, Appendix A). Next, we tested all the possible changes of the highest spliceogenic interval, c.210_219. We selected the six specific nucleotides, i.e., c.212–217, within this segment that were not shared with the overlapping microdeletions c.202_211del and c.218_227del (see Figure 2A). BRIDGES variants c.213C>T, c.214T>C, c.216C>T and c.217G>A of this segment had been already tested in the preliminary SRE assays (Table 2). Next, the 14 remaining possible nucleotide substitutions in segment c.212–c.217 were introduced into the minigene and evaluated. They represented the so-called “artificial” (ad hoc) variants that had not been identified in BRIDGES subjects. Eleven artificial variants within this interval impaired splicing, reducing the FL transcript by ≥10%, inducing the aberrant transcript ∆(E3) and increasing the relative proportion of the alternative isoform ∆(E3_5). Putting together the artificial and BRIDGES variants of c.212–217, the final proportion of spliceogenic variants was 66.7% (12/18).

In order to assess the accuracy of four ESE/ESS prediction tools (Human Splicing Finder (HSF), HEXplorer, Hot-Skip and ΔtESRseq), we pooled together and analyzed the 25 BRIDGES variants and the 14 artificial ones of the c.212–217 segment (Table 2; Appendix A) [36,37]. HEXplorer showed the highest sensitivity (74.1%, 20/27) and accuracy (61.5%) but also showed the lowest specificity (33.3%). The sensitivity of the other three predictors was very low (<37%). On the other hand, Hot-Skip presented the highest specificity (83.3%), but also 19 false negative variants. Remarkably, the spliceogenic variant c.199A>G would not have been predicted by any tool. Moreover, Hot-Skip analysis of the full exon 3 sequence suggested three changes with the highest probabilities of splicing disruption: c.163C>T (detected in BRIDGES subjects), c.163C>G and c.178C>T. We tested them in our *RAD51D* minigene, where they had weak-to-moderate effects, generating different quantities of the FL isoform (38.7–62.0%; Table 2).

Summarizing, we evaluated a total of 42 candidate ESE variants (26 identified in BRIDGES subjects), 30 of which impaired *RAD51D* exon 3 splicing.

### 3.4. ACMG/AMP-Like Classification of 37 RAD51D Variants Identified in the BRIDGES Cohort

We developed ACMG/AMP-based specifications for the interpretation of RAD51D spliceogenic variants (see Appendix A). With this approach, based on minigene read-outs (PS3/BS3 codes) and gnomADv2.1 global population frequencies (PM2/BS1/BA1 codes), we classified 37 BRIDGES variants (Table 3). As expected, most variants located at canonical splice sites (9 of 11) ended up as likely pathogenic. By contrast, many of the remaining variants ended up as variants of uncertain significance (VUS), despite demonstrating an effect on splicing.

## 4. Discussion

The progress in DNA sequencing technology has resulted in the development of multi-gene panels for clinical genetic testing of diverse genetic diseases, including hereditary breast cancer. Although next-generation sequencing is a powerful tool to identify potentially deleterious variants, most variants are classified as VUS because there are no existing datasets correlating them with their impact on gene function and cancer risk [39]. The goal of the BRIDGES initiative was to establish reliable breast cancer risk estimates for 34 putative breast cancer susceptibility genes commonly included in multigene panels by sequencing more than 60,000 breast cancer cases and 53,000 controls [5]. As a partner in this project, we have carried out the most comprehensive functional evaluation of candidate splice variants of *RAD51D* performed so far.

Despite the fact that assays using patient-derived RNA are considered the most suitable strategy to evaluate splicing outcomes, minigene assays have proved to be a useful, simple and robust approach to assess potential spliceogenic variants [12,40]. This method presents several significant advantages such as (1) analysis of single allele events without the meddling of the WT allele; (2) accurate quantification of isoforms by inhibiting the nonsense-mediated decay; (3) high versatility, allowing the study of different variants with a single minigene; (4) high reproducibility of physiological and pathological splicing patterns; (5) analysis in a cell type relevant for the disease.

In relation to the latter, we have performed all analyses in breast cancer cell line MCF-7, and we have performed a sub-analysis in the TN breast cancer cell line MDA-MB-231, replicating findings (Appendix A). Current data supports a shared heritability for ER-negative (or TN) breast cancer and ovarian cancer susceptibility [3,41,42,43]. Interestingly, epidemiological and molecular evidence indicates that high-grade serous ovarian cancer arises not from ovarian epithelial cells, but from cells in the fimbriae of the fallopian tubes [44]. At any rate, the *RAD51D* alternative splicing profiles in fimbria, ovary and breast samples are similar [33], lending further support to the relevance of our analyses. As indicated, we have performed all experiments in malignant breast epithelial cell lines, despite the fact that the carcinogenic process may target the splicing machinery [45]. Yet, in a previous study, non-malignant cells (MCF-10A) demonstrated clear disadvantages such as very slow growth and high cell lethality after transfection [46]. Further, in the same study, we observed similar minigene splicing outcomes in malignant and non-malignant cell lines.

Many studies, including previous work by our group, support the ability of minigene assays to accurately reproduce splicing alterations as observed in RNA from carriers [16,24,25,47,48,49]. Unfortunately, as far as we know, none of the *RAD51D* variants here assessed have been tested previously in RNA from carriers (the only exception being a sub-optimal study performed in a c.738+1G>A carrier, see footnote to Table 1). Yet, our *RAD51D* minigene (Figure 1) mimicked several physiological alternative splicing events [33], supporting the accuracy of the approach. Of note, splicing assays performed in RNA from carriers do not necessarily provide an accurate description of the actual spliceogenic effects caused by a genetic variant, as conflicting data in carriers of the same variant do exist (e.g., BRCA2 c.7976+5G>T or PALB2 c.3113+5G>C) [17,50,51,52]. In brief, caution should be taken when interpreting the biological and clinical implications of any splicing assay, irrespective of the experimental approach. Bearing this in mind, minigenes are certainly valuable tools for initial risk assessment, particularly if RNAs from carriers are not available.

In summary, we tested 53 *RAD51D* variants (37 from BRIDGES subjects and 16 artificial variants) in the *RAD51D* minigene: 11 splice-site and 42 putative, SRE-disrupting variants. Forty-one of them (77.4%; 11 splice-site and 30 ESE/ESS variants) impaired normal splicing patterns, supporting the high frequency of this deleterious mechanism.

### 4.1. Splice-Site Variants

Historically, a splice variant was considered critical when it affected positions ±1, 2, but other nucleotides are crucial for exon recognition, as well. Here, we analyzed 11 changes of the intron/exon boundaries, comprising seven ± 1, 2 variants, as well as other changes at intron positions −3, −4, −8 and +6 and the antepenultimate exon nucleotide. Our results showed that 10 variants dramatically disrupted splicing because they showed no trace, or residual amounts, of the FL transcript. Five variants disrupted the 3’SS, and remarkably, three of them induced the use of de novo or cryptic acceptor sites. Six variants disrupted or weakened the 5′SS wherein exon skipping was the main outcome, although two of them also provoked the use of alternative donors. Only variant c.263+6T>C showed a significant proportion of the FL transcript (49%). Finally, it is worthy of mention that MES correctly anticipated disruptions of the canonical splice sites.

### 4.2. SRE-Spliceogenic Variants

Alternative splicing is a major mechanism for increasing gene expression complexity, producing multiple mRNA and protein isoforms, and is regulated by a huge array of factors [53]. *RAD51D* exon 3 is a non-constitutive exon [33], so it is conceivable that it is controlled by ESE/ESS elements [54]. Given the low accuracy of ESE/ESS predictors, we mapped exon 3 by overlapping exonic microdeletions. This approach revealed the existence of ESE motifs throughout this exon because all the microdeletions influenced exon 3 recognition, being particularly significant at interval c.202_235. This result confirmed that this region was characterized by a high density of cis-regulatory motifs that controlled exon 3 inclusion. Fine mapping using internal 10/12-bp deletions showed that interval c.210_219 was crucial in exon 3 recognition because its deletion strongly triggered exon skipping and did not show any trace of FL transcript. Consequently, potential spliceogenic ESE variants might have appeared in these ESE-rich segments. Altogether, we tested 42 potential ESE/ESS variants mapping to these ESE-rich intervals. Of note, in a region with high levels of naturally occurring alternative splicing and most variants having limited impacts, the definition of a spliceogenic variant is controversial. In this study, we defined an arbitrary 10% reduction cutoff for the contribution of the FL transcript to the overall expression (i.e., variants with 65.8% to 73.1% of the FL transcript were not considered spliceogenic). Based on this cut-off criteria, 30 variants (71.4%; Table 2) impaired splicing, thus supporting our SRE-screening strategy. All spliceogenic SRE-variants but one (c.202G>A) produced non-negligible levels of FL transcripts (65.1% to 27.4%), with variants c.216C>G, c.215A>C, c.200_218del and c.187G>C producing 36.4%, 28.7%, 28.7% and 27.4%, respectively. c.202G>A was the only tested SRE variant that did not generate FL transcripts. Curiously, c.214T>C presented the opposite effect, strengthening exon 3 inclusion to produce only FL transcripts.

The partial impact of these variants might have been due to the disruption of independent active ESEs, which, in physiological conditions, might synergistically cooperate for efficient exon recognition, as previously reported [27]. In effect, some deletions such as c.202_235del and c.200_219del (Appendix A), which remove the crucial c.212_217 hotspot, display strong effects (no trace of the FL transcript) because they would simultaneously remove several active ESEs. In fact, any variant of the interval c.202_235 should be considered as potentially spliceogenic and functionally assayed. For instance, the recognition of atypical GC donors, which are usually associated with alternative splicing, requires the binding of several SR proteins such as SC35, Tra2β, 9G8, SF2/ASF and SRp40 [24,48,55]. Alternatively, there have been reported bi-functional SREs with enhancer and silencer properties such as the composite exonic splicing regulatory elements (CERES) [56,57], wherein the splicing impact is extremely difficult to predict. Exceptionally, one variant can decrease and increase in parallel the enhancer and silencer strengths, respectively, resulting in complete splicing anomalies. This is the case in the synonymous variant c.891C>T of the *SPINK5* gene (MIM #605010), which induces exon 11 skipping. This change reduces the interaction with enhancing factor Tra2β and concurrently increases the binding of repressor hnRNPA1 [58]. Hence, *RAD51D* c.202G>A, which did not generate any FL transcripts, in contrast to other close variants (i.e., c.198G>T or c.208G>A), might behave like this *SPINK5* variant. Likewise, the spliceogenic variants that we report here might differently alter the ESE/ESS balance such that they would induce different impacts. Nevertheless, the participation of any activator/repressor splicing factors should be demonstrated by RNA binding assays.

With regard to the ESE/ESS prediction approaches, all of them showed advantages and disadvantages (Table 2; Appendix A). HEXplorer showed the highest sensitivity (74.1%) and accuracy (61.5%) but the lowest specificity (33.3%); in fact, HEXplorer would have selected most variants of this critical segment (28 variants). However, the estimations of HSF, HEXplorer, Hot-Skip and ΔtESRseq would have failed in detecting 19, 7, 19 and 17 spliceogenic variants, respectively. For example, c.184T>A, which presented a moderately strong impact (62.8% of abnormal transcripts), would only have been detected by the less sensitive program (HSF, 29.6%). Tubeuf et al. [59] evaluated four ESE/ESS predictors, including ΔtESRseq and HEXplorer, in a large dataset of >1300 variants, wherein they found greater accuracy of both tools, HEXplorer specificity and ΔtESRseq sensitivity than in our study. This may be due to the small size of our sample or specific exon/gene parameters that may affect the performance of SRE predictors, as indicated by these authors. Nevertheless, the low accuracy of ESE/ESS prediction tools lays bare the need to improve the detection algorithms. The cumulative knowledge of the splicing regulation and functional tests could provide a basis for increasing the precision of SRE algorithms and the prediction of variant impacts. On the other hand, ESE functional mapping by exonic microdeletions appears as an optimal and reasonable strategy for discovering SRE hotspots [25,27]. Consequently, we propose to test all the reported variants that are located in these specific, highly spliceogenic intervals. Furthermore, the identification of spliceogenic ESE variants should be focused on exons that are targeted by prevalent alternative events.

### 4.3. ACMG/AMP-Based Classification of Spliceogenic Variants

Because *RAD51D* expert panel specifications of the ACMG/AMP guidelines were not yet available (clinicalgenome.org/, last accessed on 29 December 2020), we developed our own tentative specifications (Appendix A). Upon doing that, we faced several challenges, including: (i) combining predictive and functional splicing evidence, (ii) assigning specific PS3/BS3 code strengths to complex readouts (i.e., two or more mRNA transcripts with different predicted impacts on the coding sequence), and (iii) integrating a high level of naturally occurring alternative splicing into the classification system, which is of particular relevance when evaluating the pathogenicity of VUS [60]. As far as we know, no specific ClinGen-SVI recommendations addressing these issues have been yet released [31].

We tackled them by incorporating into the classification system some ad hoc rules based on structural/functional predictions and expert judgment (e.g., when assigning a PS3 code strength to complex splicing readouts, we ignored transcripts representing <10% of the overall expression). Therefore, we do not intend to provide here (see Table 3) a definitive clinical classification of genetic variants, but rather to highlight some limitations of the current ACMG/AMP framework to interpret complex spliceogenic outcomes (as in those observed for several *RAD51D* variants, see Table 1 and Table 2).

However, we think that, overall, our approach produces meaningful classifications, with spliceogenic variants ending up as variants of uncertain significance if expressing either a significant proportion of full-length transcripts (e.g., c.263+6T>C, c.185C>T) or in-frame transcripts of unknown impact on protein function (e.g., c.476_480+1dup, c.481-8C>A).

*RAD51D* c.202G>A is remarkable in that, contrary to most exon 3 variants, full-length transcripts are not produced. Yet, we have ended up classifying this variant as of uncertain significance (rather than likely pathogenic) based on: (i) the minigene readout showing a significant proportion of in-frame Δ(E3p36) transcripts, and (ii) the variant being detected in control samples (Table 3). Interestingly, *RAD51D* c.214T>C produced only full-length transcripts (therefore, suppressing the level of alternative splicing observed in wt alleles). The clinical relevance, if any, of suppressing naturally occurring alternative splicing, is far from obvious. At any rate, canonical transcripts code for a rare missense change (p.Tyr72His) that warrants classification as of uncertain significance. Indeed, for this and other RAD51D variants predicted to introduce missense changes and/or in-frame alterations, additional analyses are required to assess consequences for gene function.

Current ACMG/AMP guidelines [29] have not been developed to identify “intermediate risk variants”. Yet, we think it is worth considering this possibility for certain variants expressing variable proportions of (likely) functional and (likely) non-functional mRNAs (see Table 3 and Appendix A). It will be extremely challenging to evaluate risk for individual *RAD51D* variants that are exceedingly rare in the population, but it should be possible, in principle, to design burden analyses for variants displaying similar splicing effects.

## 5. Conclusions

We demonstrated through minigene-based analysis that splicing disruptions were induced by 28 out of 37 *RAD51D* variants (75.7%) originally identified in breast cancer cases and/or healthy controls of the BRIDGES cohort. Functional mapping by exonic microdeletions was a reliable strategy to identify SRE variants that impaired splicing. Indeed, using this approach, we identified RAD51D exon 3 sequences in which any genetic variant (regardless of its predicted coding impact) was a candidate spliceogenic variant. Finally, we have shown that current ACMG/AMP guidelines raise several issues when applied to variants associated with complex spliceogenic alterations.

## Figures and Tables

**Figure 1 cancers-13-02845-f001:**
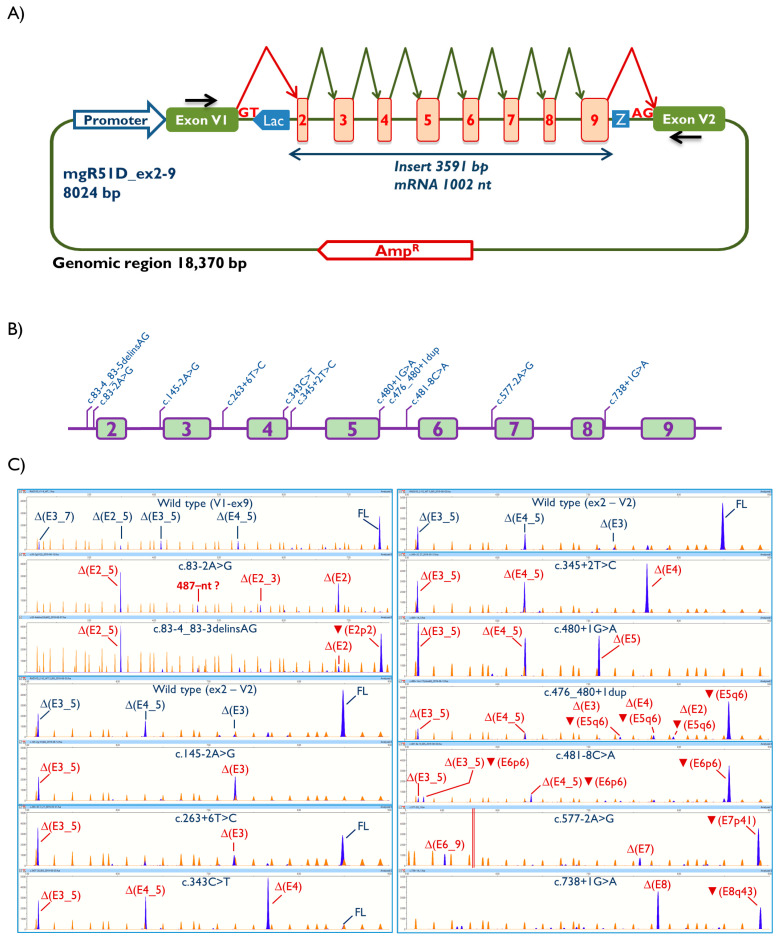
Structure of the minigene mgR51D_ex2-9 and functional assays of splice-site RAD51D variants. (**A**) Schematic representation of the RAD51D minigene, with exons 2 to 9. (**B**) Map of variants. (**C**) Fluorescent fragment analysis of transcripts generated by the wild type and mutant minigenes. FAM-labeled products (blue peaks) were run with LIZ-1200 (orange peaks) as size standard. FL: full-length transcript.

**Figure 2 cancers-13-02845-f002:**
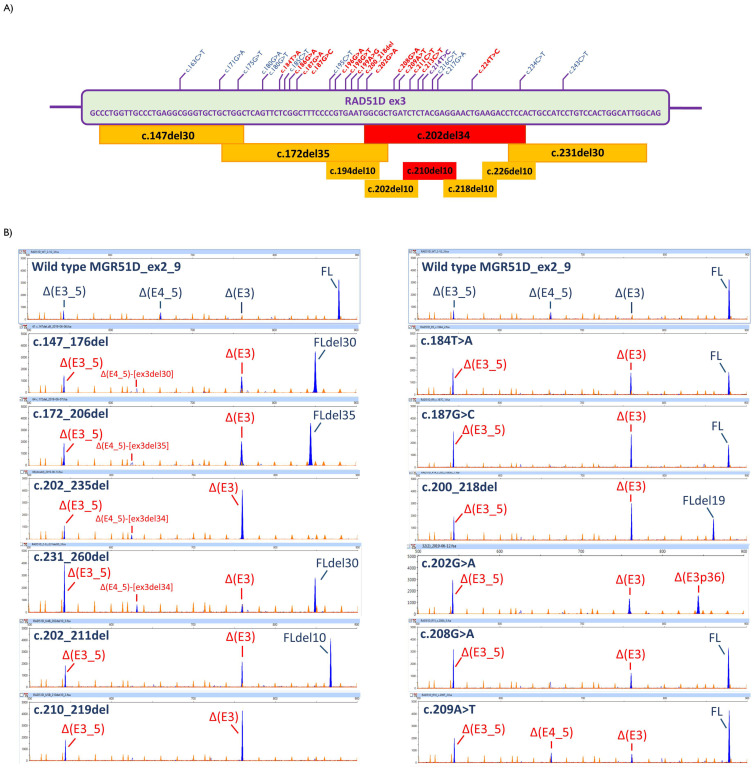
Analysis of *RAD51D* exon 3 variants. (**A**) Map of exon 3 microdeletions and tested variants. Boxes: red, total splicing disruptions; yellow, weak to moderate disruptions. Spliceogenic variants are shown in red. (**B**) Fluorescent fragment analysis of transcripts generated by selected microdeletions (left) and variants (right). FAM-labeled products (blue peaks) were run with LIZ-1200 (orange peaks) as the size standard.

**Table 1 cancers-13-02845-t001:** Bioinformatics analysis and splicing outcomes of *RAD51D* canonical splice variants.

Variant (HGVS) ^1^	Bioinformatics (MaxEnt Scan) ^2^	Transcripts ^3^			
		Canonical	PTC	In-Frame	Uncharacterized
**mgR51D_ex2-9**	Primers V1-ex9				
**Wild type**	-	57.8% ± 5.6	∆(E4_5) (13.8% ± 1.3); ∆(E3_7) (9.9% ± 2.6); ∆(E2_5) (5.2% ± 0.8)	∆(E3_5) (13.3% ± 2.4)	-
**c.83-2A>G**	[−] 3′SS (8.52→0.56)	-	∆(E2_5) (41.9% ± 1.1); ∆(E2) (39.6% ± 0.9); ∆(E2_3) (9.6% ± 0.1)	-	487-nt (9.0% ± 0.2)
**c.83-4_83-3delinsAG**	[−] 3′SS [+] 3′SS (5.42) 2-nt upstream	-	▼(E2p2) (54.2% ± 2.0); ∆(E2_5) (38.6% ± 2.1); ∆(E2) (7.2% ± 0.4)	-	-
**mgR51D_ex2-9**	Primers ex2-V2			-	-
**Wild type**		73.1% ± 5.6	∆(E4_5) (9.4% ± 3)	∆(E3_5) (17.5% ± 5.2)	-
**c.145-2A>G**	[−] 3′SS (2.43)	-	∆(E3) (55.5% ± 0.6)	∆(E3_5) (44.5% ± 0.6)	-
**c.263+6T>C**	5′SS: 7.44→4.86	49.0% ± 1.2	∆(E3) (15.6% ± 0.3)	∆(E3_5) (35.4% ± 1.2)	-
**c.343C>T**	5′SS: 7.79→4.36	3.4% ± 0.7	∆(E4) (45.7% ± 1.0); ∆(E4_5) (28.1% ± 0.6)	∆(E3_5) (22.7% ± 1.2)	-
**c.345+2T>C**	[−] 5′SS (7.79→0.04)	-	∆(E4) (49.6% ± 1.4); ∆(E4_5) (26.4% ± 1.0)	∆(E3_5) (24.0% ± 0.7)	-
**c.480+1G>A**	[−] 5′SS (11.08→2.9)	-	∆(E4_5) (29.2% ± 0.4)	∆(E3_5) (41.2% ± 0.7); ∆(E5) (29.6% ± 0.4)	-
**c.476_480+1dup**	[−] 5′SS (11.08→0.5)	-	∆(E4_5) (8.6% ± 0.4); (∆(E4) ▼(E5q6)) (6.3% ± 0.1); (∆(E2) ▼(E5q6)) (4.3% ± 0.1); (∆(E3) ▼(E5q6)) (3.6% ± 0.4)	▼(E5q6) (60.4% ± 4.3); ∆(E3_5) (16.7% ± 5.0)	-
**c.481-8C>A**	3′SS: 8.21→1.75 [+] 3′SS (11.06) 6-nt upstream	0.4% ± 0.1	(∆(E4_5) ▼(E6p6)) (13.4% ± 0.2)	▼(E6p6) (70.0% ± 0.8); (∆(E3_5) ▼(E6p6)) (8.8% ± 0.3); ∆(E3_5) (7.5% ± 0.3)	-
**c.577-2A>G**	[−] 3′SS (10.36→2.41)	-	▼(E7p41) (65.5% ± 0.5); ∆(E7) (14.3% ± 0.2)	∆(E6_9) (20.2% ± 0.8)	-
**c.738+1G>A ^4^**	[−] 5′SS (6.13→−2.05)	-	▼(E8q43) (27.9% ± 0.1); ∆(E8) (51.1% ± 0.2); ∆(E4_7) (10.9% ± 0.1)	-	1363-nt (10.1% ± 0.0)

^1^ Variants without full-length transcripts or residual amounts (<5%) are indicated in bold font. ^2^ [−]: site disruption; [+]: new site. ^3^ Transcripts are described with a combination of the following symbols: ▼ (incorporation of intronic sequences that were not present in the reference transcript), ∆ (skipping of exonic sequences that were present in the reference transcript), E (exon), p (new acceptor site), q (new donor site) and a number representing the exact number of nucleotides incorporated or skipped. For example, ▼(E2p2) denotes the use of an alternative acceptor site 2 nucleotides upstream of exon 2, causing the addition of 2-nt to the mature mRNA. ^4^ According to reference [33], c.738+1G>A causes only a 37-nucleotide intron retention ▼(E8q37). Yet, some methodological issues (location of PCR primers not reported, RT-PCR products analyzed only by low-sensitivity agarose electrophoresis, no agarose band isolation before Sanger sequencing and no allele-specific information) precluded, in our opinion, direct comparison with our minigene results.

**Table 2 cancers-13-02845-t002:** Bioinformatics analysis and splicing outcomes of putative exon 3 ESE variants.

Variant ^1^	Transcripts ^2^			In Silico Tools ^3^			
	Canonical	PTC	In-Frame	HSF	HEXplorer	Hot-Skip	ΔtESRseq
**Wild type**	73.1% ± 5.6	∆(E4_5) (9.4% ± 3)	∆(E3_5) (17.5% ± 5.2)	-			
**BRIDGES variants**							
**c.171G>A**	76.0% ± 1.4	∆(E4_5) (8.4% ± 0.4)	∆(E3_5) (18.6% ± 1.1)	-	**−40.3**	0	**−1.03**
**c.175G>T**	**52.9% ± 2.6**	∆(E4_5) (9.7% ± 1.1) **∆(E3)** (9.6% ± 0.4)	**∆(E3_5)** (27.7% ± 1.2)	-	**−27.0**	**2**	1.30
**c.180G>A**	67.5% ± 2.6	∆(E4_5) (14.4% ± 1.1)	∆(E3_5) (18.1% ± 1.5)	-	2.5	0	0.25
**c.180G>T**	69.4% ± 0.5	∆(E4_5) (13.7% ± 0.1)	∆(E3_5) (16.9% ± 0.4)	-	**−65.1**	**5**	0.25
**c.184T>A**	**37.2% ± 0.6**	**∆(E3)** (31.1% ± 1.5)	**∆(E3_5)** (31.7% ± 1.5)	**+**	23.3	0	−0.67
**c.185C>T**	66.3% ± 1.2	∆(E4_5) (9.9% ± 0.5)	∆(E3_5) (23.8% ± 1.5)	-	**−53.9**	**7**	**−1.85**
**c.186G>A**	**64.3% ± 0.5**	∆(E4_5) (7.4% ± 0.1) **∆(E3)** (8.5% ± 0.0)	∆(E3_5) (19.7% ± 0.4)	**+**	**−23.2**	0	**−1.88**
**c.187G>A**	**59.8% ± 2.9**	**∆(E3)** (13.9% ± 0.9)	**∆(E3_5)** (26.3% ± 2.3)	**+**	13.4	0.3	0.65
**c.187G>C**	**27.4% ± 0.4**	**∆(E3)** (38.3% ± 0.4)	**∆(E3_5)** (34.2% ± 0.3)	-	**−12.4**	**1**	−0.05
**c.195C>T**	69.7% ± 0.3	∆(E4_5) (10.2% ± 0.1)	∆(E3_5) (20.1% ± 0.3)	-	**−22.8**	0.4	−0.26
**c.196G>A**	**58.8% ± 1.6**	∆(E4_5) (5.4% ± 0.3) **∆(E3)** (10.6% ± 0.8)	**∆(E3_5)** (33.3% ± 2.1)	-	**−17.2**	0	**−1.72**
**c.198G>T**	**48.7% ± 1.8**	∆(E4_5) (5.9% ± 0.3) **∆(E3)** (17.1% ± 0.7)	**∆(E3_5)** (28.3% ± 0.9)	-	**−47.3**	**5**	**−2.69**
**c.199A>G**	**65.1% ± 1.0**		**∆(E3_5)** (34.9% ± 1.0)	-	37.8	0.7	1.31
**c.200_218del**	**28.7% ± 0.5**	**∆(E3)** (46.9% ± 0.0)	**∆(E3_5)** (24.4% ± 0.5)	-	**−119.8**	0.26	**−4.14**
**c.202G>A**	**-**	**∆(E3)** (26.7% ± 0.9)	**∆(E3_5)** (41.1% ± 3.1); **∆(E3p36)** (32.2% ± 2.9)	-	**−38.5**	0	**−1.58**
**c.208G>A**	**46.8% ± 1.1**	**∆(E3)** (16.8% ± 0.2)	**∆(E3_5)** (36.4% ± 1.2)	**+**	**−52.1**	0	**−1.37**
**c.209A>T**	**58.2% ± 1.0**	∆(E4_5) (10.0% ± 0.3) **∆(E3)** (9.2% ± 0.2)	∆(E3_5) (22.6% ± 0.9)	-	**−58.0**	**5**	−0.30
**c.211C>T**	**60.6% ± 0.7**	∆(E4_5) (8.6% ± 0.0) **∆(E3)** (7.9% ± 0.3)	**∆(E3_5)** (23.0% ± 0.5)	-	**−63.2**	**2.5**	0.19
**c.213C>T**	**57.2% ± 0.4**	**∆(E3)** (19.5% ± 0.0)	**∆(E3_5)** (23.3% ± 0.3)	-	**−53.1**	**4**	**−1.55**
**c.214T>C**	100.0% ± 0.0			**+**	6.7	0	−0.30
**c.216C>T**	68.4% ± 0.2	**∆(E3)** (14.0% ± 0.0)	∆(E3_5) (17.6% ± 0.4)	-	**−34.1**	0	**−2.38**
**c.217G>A**	66.5% ± 2.0	∆(E4_5) (12.5% ± 0.3)	∆(E3_5) (21.0% ± 1.7)	**+**	**−7.8**	0	**−0.94**
**c.224T>C**	**59.9% ± 0.9**	∆(E4_5) (13.4% ± 0.1)	**∆(E3_5)** (26.7% ± 0.8)	-	**−9.3**	0	0.27
**c.234C>T**	70.7% ± 0.5	∆(E4_5) (11.5% ± 0.2)	∆(E3_5) (17.8% ± 0.4)	**+**	−0.3	0	0.10
**c.243C>T**	**50.9% ± 1.2**	**∆(E3)** (16.8% ± 0.1)	**∆(E3_5)** (32.3% ± 1.1)	-	**−59.7**	**4**	−0.58
**Artificial Variants c.212_217**							
**c.212T>A**	**63.6% ± 2.3**	**∆(E3)** (5.5% ± 0.3) ∆(E4_5) (5.5% ± 0.0)	**∆(E3_5)** (25.4% ± 2.6)	**+**	16.1	0	0.51
**c.212T>C**	**62.7% ± 1.0**	**∆(E3)** (5.5% ± 0.1) ∆(E4_5) (7.1% ± 0.0)	**∆(E3_5)** (24.7% ± 0.9)	**+**	−0.4	0	−0.01
**c.212T>G**	**63.4% ± 2.1**	**∆(E3)** (5.9% ± 0.2) ∆(E4_5) (4.9% ± 0.2)	**∆(E3_5)** (25.8% ± 1.9)	**+**	41.8	0	1.58
**c.213C>A**	**41.4% ± 0.3**	**∆(E3)** (11.1% ± 0.2) ∆(E4_5) (6.4% ± 0.0)	**∆(E3_5)** (41.1% ± 0.5)	-	**−25.6**	0.5	**−0.81**
**c.213C>G**	**49.6% ± 0.6**	**∆(E3)** (6.7% ± 0.1) ∆(E4_5) (6.1% ± 0.0)	**∆(E3_5)** (37.6% ± 0.7)	-	**−20.1**	0.5	0.57
**c.214T>A**	**64.3% ± 1.3**	∆(E4_5) (10.9% ± 0.1)	**∆(E3_5)** (24.8% ± 1.3)	**+**	19.1	0	0.87
**c.214T>G**	65.8% ± 2.4	∆(E4_5) (11.3% ± 0.3)	∆(E3_5) (18.9% ± 0.4)	-	18.3	0	3.41
**c.215A>C**	**28.7% ± 0.2**	∆(E4_5) (9.6% ± 0.3)	**∆(E3_5)** (61.7% ± 0.4)	-	**−40.5**	0	1.82
**c.215A>G**	69.9% ± 1.6	∆(E4_5) (9.4% ± 0.2)	∆(E3_5) (20.7% ± 1.5)	-	**−27.9**	0.33	1.78
**c.215A>T**	70.6% ± 1.7	∆(E4_5) (8.7% ± 0.1)	∆(E3_5) (20.7% ± 1.6)	-	**−21.2**	0.67	1.90
**c.216C>A**	**54.7% ± 1.0**	**∆(E3)** (12.8% ± 0.4)	**∆(E3_5)** (32.5% ± 1.4)	-	**−73.7**	0.13	−0.56
**c.216C>G**	**36.4% ± 0.6**	**∆(E3)** (22.1% ± 1.3)	**∆(E3_5)** (41.6% ± 1.9)	-	**−91**	0.33	**−1.15**
**c.217G>C**	**62.7% ± 0.5**	**∆(E3)** (8.3% ± 0.1)	**∆(E3_5)** (29.0% ± 0.4)	-	**−21.2**	0	−0.27
**c.217G>T**	**63.6% ± 0.4**	**∆(E3)** (14.3% ± 0.3)	∆(E3_5) (22.1% ± 0.5)	-	**−114.2**	**1.67**	**−1.66**
**Hot-Skip Variants**							
**c.163C>G**	**38.7% ± 3.0**	**∆(E3)** (26.9% ± 0.7)	**∆(E3_5)** (34.4% ± 2.5)	-	**−92.7**	**18**	**−2.81**
**c.163C>T** **(BRIDGES)**	**62.0% ± 0.5**	∆(E4_5) (12.3% ± 0.3) **∆(E3)** (5.6% ± 0.1)	∆(E3_5) (20.1% ± 0.6)	- MES: new 5′SS (7.07)	**−52.0**	**16**	**−1.978**
**c.178C>T**	**54.3% ± 1.0**	**∆(E3)** (16.0% ± 0.1); ∆(E4_5) (4.8% ± 0.1)	**∆(E3_5)** (24.9% ± 0.8)	-	**−44.7**	**12**	**−2.08**

^1^ Spliceogenic variants are indicated in bold type. ^2^ Remarkable transcript variations are indicated in bold type: ≥10% reduction of the canonical transcript, detection of the anomalous ∆(E3) or increase of the ∆(E3_5) isoform higher than 22.7% (mean + s.d. of ∆(E3_5) in the wt minigene). ^3^ Bold font: HSF, alteration of the ESE/ESS motifs ratio [+]; scores of HEXplorer, Hot-Skip and ΔtESRseq below or above their corresponding cutoffs (≤−5, ≥1 and ≤−0.75, respectively).

**Table 3 cancers-13-02845-t003:** Proposed pSAD-based ACMG/AMP clinical classification of 37 *RAD51D* genetic variants detected in the BRIDGES cohort.

c.HGVS ^1^	p.HGVS ^1^	Clinvar ^2^	PS3/BS3 ^3^	PM2/BS1/BA1 ^4^	Proxy for Allele Counts ^5^	Variant Classification ^6^
c.83-4_-3delinsAG	p.?	VUS (**)	PS3_VS	(0/251433) PM2	rs780590372 (=)	Likely Pathogenic (PS3_VS + PM2)
c.83-2A>G	p.?	LP (*)	PS3_VS (91%PS3_VS + 9%N/A) ^7^	(0/251433) PM2	rs780590372 (−1)	Likely Pathogenic (PS3_VS + PM2)
c.145-2A>G	p.?	not reported	PS3_VS	(0/251102) PM2	rs201974522 (−1)	Likely Pathogenic (PS3_VS + PM2)
c.163C>T	p.(Arg55Trp)	VUS (**)	N/A (62%N/A + 38%PS3_VS)	(2/251433) PM2	-	Uncertain Significance (PM2 only) ^8^
c.171G>A	p.(Leu57=)	LB (**)	BS3 (76%BS3 + 24%PS3_VS)	(0/249480) PM2	rs745307359 (−4)	Uncertain Significance (BS3 + PM2)
c.175G>T	p.(Ala59Ser)	not reported	N/A (53%N/A + 47%PS·_VS)	(0/251298) PM2	rs780689600 (+4)	Uncertain Significance (PM2 only) ^8^
c.180G>A	p.(Gln60=)	LB (*)	BS3 (68%BS3 + 32%PS3_VS)	(0/251298) PM2	rs780689600 (−1)	Uncertain Significance (BS3 + PM2)
c.180G>T	p.(Gln60His)	VUS (**)	N/A (69%N/A + 31%PS3_VS)	(0/251298) PM2		Uncertain Significance (PM2 only) ^8^
c.184T>A	p.(Ser62Thr)	not reported	N/A (37%N/A + 63%PS3_VS)	(0/251382) PM2	rs374357106 (+3)	Uncertain Significance (PM2 only) ^8^
c.185C>T	p.(Ser62Leu)	VUS (**)	N/A (66%N/A + 34%PS3_VS)	(5/251382) N/A	-	Uncertain Significance (no codes) ^8^
c.186G>A	p.(Ser62=)	LB (**)	N/A (64%BS3 + 36%PS3_VS)	(3/251380) N/A	-	Uncertain Significance (no codes) ^8^
c.187G>A	p.(Ala63Thr)	VUS (**)	N/A (60%N/A + 40%PS3_VS)	(1/251408) PM2	-	Uncertain Significance (PM2 only) ^8^
c.187G>C	p.(Ala63Pro)	not reported	N/A (27%N/A + 73%PS3_VS)	(0/251408) PM2	c.187G>A	Uncertain Significance (PM2 only) ^8^
c.195C>T	p.(Pro65=)	LB (**)	BS3 (70%BS3 + 30%PS3_VS)	(5/251420) N/A	-	Uncertain Significance (BS3 only)
c.196G>A	p.(Val66Met)	B (2), LB (4), VUS (4)	N/A (59%N/A + 41%PS3_VS)	(80/251384) BS1	-	Uncertain Significance (BS1) ^8^
c.198G>T	p.(Val66=)	LB (**)	N/A (49%BS3 + 51%PS3_VS)	(9/251448) N/A	-	Uncertain Significance (no codes) ^8^
c.199A>G	p.(Asn67Asp)	not reported	N/A (65%N/A + 35%PS3_VS)	(0/251448) PM2	rs546461804 (+1)	Uncertain Significance (PM2 only) ^8^
c.200_218del	p.(Asn67Argfs*7)	not reported	PS3_VS	(0/251448) PM2	rs54661804 (=)	Likely Pathogenic (PS3_VS + PM2)
c.202G>A	p.(Gly68Ser)	VUS (**)	PS3 (68%PS3_VS + 32%PS3)	(9/251454) N/A	-	Uncertain Significance (PS3 only)
c.208G>A	p.(Asp70Asn)	VUS (**)	N/A (47%N/A + 53%PS3_VS)	(8/251458) N/A	-	Uncertain Significance (no codes) ^8^
c.209A>T	p.(Asp70Val)	VUS (**)	N/A (58%N/A + 42%PS3_VS)	(1/251450) PM2	-	Uncertain Significance (PM2 only) ^8^
c.211C>T	p.(Leu71Phe)	VUS (*)	N/A (61%N/A + 39%PS3_VS)	(0/251476) PM2	rs559850711 (+1)	Uncertain Significance (PM2 only) ^8^
c.213C>T	p.(Leu71=)	LB (**)	N/A (57%BS3 + 43%PS3_VS)	(2/251466) PM2	-	Uncertain Significance (PM2 only) ^8^
c.214T>C	p.(Tyr72His)	not reported	N/A	(0/251466) PM2	rs745546403 (−1)	Uncertain Significance (PM2 only)
c.216C>T	p.(Tyr72=)	B/LB (**)	BS3 (67%BS3 + 33%PS3_VS)	(27/251474) N/A	-	Uncertain Significance (BS3)
c.217G>A	p.(Glu73Lys)	VUS (**)	N/A (67%N/A + 33%PS3_VS)	(2/251462) PM2	-	Uncertain Significance (PM2 only) ^8^
c.224T>C	p.(Leu75Pro)	not reported	N/A (60%N/A + 40%PS3_VS)	(0/251466) PM2	rs746929682 (−1)	Uncertain Significance (PM2 only) ^8^
c.234C>T	p.(Ser78=)	B (**)	BS3 (71%BS3 + 29%PS3_VS)	(29273/251448) BA1	-	Benign (BS3 + BA1)
c.243C>T	p.(Ile81=)	LB (**)	N/A (60%BS3 + 40%PS3_VS)	(1/251472) PM2	-	Uncertain Significance (PM2 only) ^8^
c.263 + 6T>C	p.?	not reported	N/A (49%BS3 + 51%PS3_VS)	(0/251424) PM2	rs56218020 (+1)	Uncertain Significance (PM2) ^8^
c.343C>T	p.(Gln115Ter)	P (**)	PS3_VS	(0/141295) PM2	rs786202507 (−4)rs878854562(+7)	Likely Pathogenic (PS3_VS + PM2)
c.345+2T>C	p.?	LP(1); VUS(1)	PS3_VS	(0/251220) PM2	rs878854562 (+3)	Likely Pathogenic (PS3_VS + PM2)
c.476_480+1dup	p.?	not reported	N/A (40%PS3_VS + 60%N/A)	(0/251474) PM2	rs1057521922 (=)	Uncertain Significance (PM2 only) ^8^
c.480+1G>A	p.?	(-)	PS3 (70%PS3_VS + 30%PS3)	(0/251474) PM2	rs1057521922 (−3)	Likely Pathogenic (PS3 + PM2)
c.481-8C>A	p.?	not reported	N/A (30%PS3_VS + 70%N/A)	(0/241990) PM2	rs762247126 (=)	Uncertain Significance (PM2 only) ^8^
c.577-2A>G	p.?	P/LP (**)	PS3 (80%PS3_VS + 20%PS3)	(0/250980) PM2	rs1210749655 (−4)	Likely Pathogenic (PS3 + PM2)
c.738+1G>A	p.?	LP (**)	PS3_VS (70%PS3_VS + 10%N/A) ^7^	(0/240992) PM2	rs1210620444 (−1)	Likely Pathogenic (PS3 + PM2) ^9^

^1^ NM_002878.3. ^2^ ClinVar last accessed 10 May 2021. VUS: variant of uncertain significance, LP: likely pathogenic, P: pathogenic, LB: likely benign, B: benign. Criteria provided + multiple submitters + no conflicts (**), criteria provided + single submitter (*), no assertion criteria provided (-). If interpretations conflict, the number of submitters supporting each interpretation is indicated. ^3^ pSAD read-outs were deconvoluted into individual transcripts that were interpreted as per ClinGen-SVI PVS1 decision tree recommendations; very strong pathogenic: PS3_VS; strong pathogenic: PS3; strong benign: BS3; BA1: stand-alone benign, allele frequency >5%; N/A: no code strengths applicable. If transcripts with different evidence strengths were observed, the approximated contribution to the overall expression is shown (%). Combined PS3 strength was based on expert judgment (see Section 2). ^4^ For rarity evidence, we counted alleles in gnomADv2.1 global. ^5^ If absent from gnomADv2.1 global, counted alleles were inferred using the nearest reference SNP as a proxy (−/+, upstream/downstream distance, = overlapping). ^6^ Not intended as a definitive clinical classification (see Section 4). ^7^ Both variants expressed a non-negligible proportion of transcripts not fully characterized (see Table 1 and Appendix A). ^8^ Candidate intermediate risk variants (see Section 4). ^9^ Despite its differences, both the current study and references [38] support a likely pathogenic classification for this variant.

## Data Availability

After publication, all sequencing and fragment analysis data will be available at Figshare (DOI: 10.6084/m9.figshare.10272029).

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
