# Peer review of "RAD51D* Aberrant Splicing in Breast Cancer: Identification of Splicing Regulatory Elements and Minigene-Based Evaluation of 53 DNA Variants"

_cancers, 2021, doi:10.3390/cancers13112845_

Round 1
Reviewer 1 Report
Summary
In the manuscript entitled “RAD51D aberrant splicing in breast cancer: identification of splicing regulatory elements and functional evaluation of 53 DNA variants”, Bueno-Martinez et al generated a minigene of RAD51D exon2-9 in which, they introduced various mutations of splice sites identified through a BRIDGES cohort of breast/ovarian cancer related genomic variant. The constructs were tested and the production of RAD51D variants analysed in MCF7 breast cancer cells, with an emphasis on minigenes containing mutation in the alternative exon 3. Overall, 41 spliceogenic variants were identified within 9 of them most likely producing pathogenic variants.
Strong point
Impact on clinical analysis and therapy prevention possibilities
Minigene system is elegantly used and the study well conducted
The supplementary method is very well described, allowing a better understanding of the manuscript workflow
Weak point
The minigene system is only tested in one cell types.
The link between pathological behaviour and variants/spliceogenic alteration could be re-enforced by testing some of these minigenes in other cancer or normal cell contexts.
The results will gained visibility by adding a small sentence of conclusion for each paragraph
Major comments
-Variant leading to less than 10% reduction are considered weak. However, if I understand well, the NMD machinery is inhibited. Thus, if these mutation lead to a higher degradation of the RNA, it will not be detected while it may have a strong relevance clinically. How the author envision this point ? Could this weak mutations be tested without repressing NMD pathway? Would it be relevant despite that the minigene is not the full transcript?
-How about more aggressive breast cancer line such as the triple negative MDA-MB-231 cell line? It would re-inforce the impact of the study to test some of the minigenes in this cell line.
-Moreover, it exists non-cancerous breast cancer cells (MCF10A). What is the response of the minigenes (at least some selected one, pathogenic and non pathogenic predicted) in this type of cell line?
-It is mentioned that RAD51D has also a strong impact on ovarian cancers, also estrogen responsive. What is the activity of the minigenes in ovary cancer cell line? Do you get the same response of the selected pathogenic variants? This could be tested too.
Minor comment
Line 35: “alsotested” miss space
Author Response
We thank the reviewer for their thorough scrutiny of this work and the helpful comments.
The line numbers where the corrections have been introduced corresponds to the revised version of the manuscript (Manuscript_RAD51D_Cancers_R1-Track_Changes.docx) in which all the changes can be tracked. All changes have been also highlighted in the final manuscript file.
Weak point
The minigene system is only tested in one cell types.
Response: See explanations below
The link between pathological behaviour and variants/spliceogenic alteration could be re-enforced by testing some of these minigenes in other cancer or normal cell contexts.
Response: See explanations below
The results will gained visibility by adding a small sentence of conclusion for each paragraph.
Response:
We have thoroughly modified Supplementary Figure S1 (graphics and legend) to clarify it.
In addition, we know that the most complex part of the manuscript is the section of ESE-variants of exon 3, so we have introduced a final summarizing sentence, and several explanatory sentences throughout this section in Results and Discussion
Major comments
-Variant leading to less than 10% reduction are considered weak. However, if I understand well, the NMD machinery is inhibited. Thus, if these mutation lead to a higher degradation of the RNA, it will not be detected while it may have a strong relevance clinically. How the author envision this point ? Could this weak mutations be tested without repressing NMD pathway? Would it be relevant despite that the minigene is not the full transcript?
Response:
*10% Cut-off
We have to highlight that the RAD51D gene is linked to high levels of alternative splicing. Likewise, the splicing pattern of the wild type minigene is very similar to the physiological one, mimicking at least five alternative events. Given that the standard deviation of the full-length transcript proportion in the wild type minigene is ±5.6 (equivalent to 7.7% increase/reduction), we were conservative and selected a 10%-reduction as the cut-off to consider that a variant affects normal splicing.
A reference to the <10% cut-off has been removed from the results section on canonical splice-site variants, as it is not relevant. All 11 splice-site variants show a reduction of the FL-transcripts contribution >>10% (i.e. no canonical splice-site variants have been considered weak).
The 10% cut-off has been only used to identify spliceogenic variants targeting exon 3 ESEs. In the revised version, the cut-off definition has been improved (Result section, lines 244-245):
“In this study, variants with a FL-transcript proportion ranging from 65.8%-73.1% (<10% reduction of the FL-transcript in the wt minigene) were not considered spliceogenic (see Discussion).”
Further, the rationale supporting the need for a cut-off has been included in the Discussion section (lines 409-418).
“Altogether, we tested 42 potential ESE/ESS variants mapping to these ESE-rich intervals. Of note, in a region with high levels of naturally occurring alternative splicing and most variants with limited impacts, it is controversial the definition of a spliceogenic variant. In this study, we have defined an arbitrary 10%-reduction cut-off for the contribution of the FL-transcript to the overall expression (i.e. variants with 65.8% to 73.1% of the FL-transcript are not considered spliceogenic). Based on this cut-off criteria, 30 variants (71.4%; Table 2) impaired splicing, thus supporting our SRE-screening strategy. All spliceogenic SRE-variants but one (c.202G>A), produced non-negligible levels of FL-transcripts (65.1% to 27.4%), with variants c.216C>G, c.215A>C, c.200_218del, and c.187G>C producing 36.4%, 28.7%, 28.7%, and 27.4%, respectively.”
*Use of NMD inhibitors:
The use of NMD inhibitors is recommended in mRNA splicing assay protocols (PMID: 24212087). As the reviewer rightly suggests, NMD biases the proportions of the different NMD and non-NMD transcripts generated by a variant in any cell type (PTC- vs. in-frame transcripts). Consequently, contribution from NMD-prone transcripts will be underestimated (i.e. contribution of non-NMD transcripts will be overestimated) and rare PTC-transcripts may be missed.
We already checked the use of NMD-inhibitors in Acedo et al (2015) and confirmed the presence of significant differences between both experimental conditions. Therefore, NMD-inhibitors are necessary to detect all the transcripts that are induced by a variant and to accurately estimate their relative amounts.
It is highly likely that without NMD inhibitors, we could not observe those weak impacts or detect rare isoforms. Furthermore, we want to underline that we also use a highly sensitive and high-resolution method: fluorescent fragment analysis electrophoresis that allows the detection of rare splicing isoforms (even <5% of the relative proportion) and can distinguish between transcripts that differ in only 1-2 nt. These data could not have been obtained if we had used other less-sensitive methods: agarose gels (low resolution and sensitivity) or sequencing (this would not have detected the less frequent transcripts). In conclusion, we think that repetition of any experiment without a NMD-inhibitor will not provide useful information as we showed in Acedo et al (2015).
More relevant, none of the exon 3 ESE BRIDGES variants tested introduces a PTC, and therefore there is no reason to suspect that NMD inhibition is masking a higher degradation of FL-RNA for any of these variants. We firmly believe that it is not necessary to carry the suggested experiments without NMD-inhibition.
*How about more aggressive breast cancer line such as the triple negative MDA-MB-231 cell line? It would re-inforce the impact of the study to test some of the minigenes in this cell line.
Response:
Acknowledge this comment.
Interestingly, according to the studies of Dorling et al (2021) in more than 113,000 women, RAD51D pathogenic variants are significantly associated with a higher risk in ER negative (2.92) and triple-negative breast cancer (6.01) vs. an odds ratio of 1.80 of overall breast cancer risk.
So, it is very interesting to use a triple negative cell line to confirm MCF-7 cells and we have repeated eight canonical splice-site variant experiments in the MDA-MB-231 cell line (as suggested by the reviewer). As expected, the splicing outcomes of minigenes in MCF-7 cells were identical or very similar to those in MDA-MB-231 cells.
We want to remark that splicing outcomes of minigenes in MCF-7 cells are highly reproducible as we have shown in several publications (38 variants showed the same splicing outcomes in patient vs. minigene RNA assays).
*Consequently, Materials and Methods and Results have been modified, and we have included a new Supplementary Figure with an agarose gel with the results of this cell line:
- Materials and Methods, lines 133-136:
“The triple negative breast cancer cell line MDA-MB-231 was cultured in DMEM medium supplemented with 10% Fetal Bovine Serum, 2 mM glutamine, 1% non-essential amino acids and 1% Penicillin/Streptomycin solution.”
- Results, lines 183-186:
“Eight of these variants (c.263+6T>C, c.343C>T, c.345+2T>C, c.480+1G>A, c.476_480+1dup, c.481-8C>A, c.577-2A>G and c.738+1G>A) were also tested in the triple-negative breast cancer cells MDA-MB-231, in which they replicated the splicing profiles (Supplementary Figure S3).”
- Discussion, lines 336-351:
“In relation with the latter, we have performed RAD51D minigene analyses in two breast cancer cell lines (MCF7 and MDA-MB-231), the latter derived from a triple-negative breast cancer (TNBC) patient. We assayed eight spliceogenic … … but not with ER-positive breast cancer risk [5,6]. Germ-line defects in all four genes are associated as well with increased ovarian cancer risk (for BARD1 the evidence is limited) [3,41-43].”
-Moreover, it exists non-cancerous breast cancer cells (MCF10A). What is the response of the minigenes (at least some selected one, pathogenic and non pathogenic predicted) in this type of cell line?
Response:
Acknowledge this comment.
Most cells used in minigene experiments are cancer cell lines (HeLa, MCF-7, HepG2, HEK 293, etc.). However, we ruled out the use of MCF10A cells for several reasons (Sanz et al 2010). First, we used MCF10A in Sanz et al (2010), and we obtained the same or similar results with several BRCA1 variants. Second, MCF10A cells grow very slowly; third, when these cells are transfected with Lipofectamine or similar reagents, cells almost disappear and even grow slower so RNA can be barely purified after 2 weeks of very slow growth (vs. 1-2 days for MCF-7, HeLa, HepG2, MDA-MB-23, etc). This would seriously delay all the experiments so that it is really complex to perform them in a short period of time, and given that they yield similar results, it is not practical to use these cells.
* We have introduced three new sentences, Discussion, lines 357-363:
“On the other hand, minigene assays are commonly performed in tumor cells (MCF-7, HeLa, HepG2, HEK 293, etc.) because they are easy to handle and generate highly repro-ducible splicing patterns. However, it is known that the splicing machinery of cancer cells may be affected [45]. In this regard, in a previous study we ruled out the use of non-tumor breast cells (MCF-10A) since the splicing outcomes were very similar to those of tumor cells, and showed clear disadvantages, such as very slow growth or high cell lethality after transfection [46].”
-It is mentioned that RAD51D has also a strong impact on ovarian cancers, also estrogen responsive. What is the activity of the minigenes in ovary cancer cell line? Do you get the same response of the selected pathogenic variants? This could be tested too.
Response:
Acknowledge this comment.
There is mounting evidence pointing towards a shared heritability for ER-negative (or TN) breast cancer and ovarian cancer susceptibility. Moreover, a comprehensive characterization of RAD51D alternative splicing profile in non-malignant breast and fimbriae tissues has discarded major tissue specific effects. Based on that, we consider plausible that minigene experiments in ovarian cancer and TNBC cell lines yield very similar results.
We also want to highlight that: i) the BRIDGES project (on which this study is based) is focused on the estimation of the breast cancer risk (study of a panel of 34 known or suspected breast cancer genes in in 60,466 breast cancer patients and 53,461 controls); ii), despite the higher relative risk of ovarian cancer, the absolute risk of RAD51D pathogenic variants is higher or similar for breast cancer than for ovarian cancer; iii) we have replicated the splicing outcomes of 8 variants in a triple-negative cell line and given the high reproducibility of minigene assays we do not expect substantial differences with an ovarian cell line. As indicated in several previous reports, there are no significant differences of the resultant splicing profiles between different cell lines (MCF-7, MCF-10A, HeLa or HepG2), where we have performed nearly 1,000 different assays in different cell lines; iv) it is worth mentioning that RT-PCR of patient RNA is assumed as the best option to test candidate splicing variants and blood is the main source of patient RNA, so that tissue-specific splicing differences could be even more evident.
Finally, no ovarian cell lines were available, so these experiments cannot be carried out in a reasonable period of time (cell line + optimization of minigene assays). In any case, we also want to underline that these minigene experiments cannot be repeated quickly since they are very laborious. However, we think that this reviewer’s suggestion is very useful for future experiments, so we will acquire a human ovarian cancer cell line to include in our battery of cell lines.
- We have included two new paragraphs (Discussion, lines 336-357) where we discuss this issue:
“In relation with the latter, we have performed RAD51D minigene analyses in two breast cancer cell lines (MCF7 and MDA-MB-231), the latter derived from a triple-negative breast cancer (TNBC) patient. … … Remarkably, a comprehensive characterization of RAD51D alternative splicing profile in non-malignant breast and fimbriae tissues has discarded major tissue specific effects [33]. Based on that, we consider plausible that minigene experiments in ovarian cancer and TNBC cell lines yield very similar results.”
Minor comment
Line 35: “alsotested” miss space
Response:
- Corrected
Reviewer 2 Report
Bueno-Martínez et al. use a minigene approach in MCF7 breast cancer cells for the evaluation of RAD51D sequence variants that are predicted to affect mRNA splicing. This is clinically important work, because pathogenic RAD51D mutations increase the risk to develop ovarian cancer and (to a lesser extent) breast cancer. Although the paper is somewhat difficult to read and interpret, the data seem solid and informative, thereby justifying publication. However, I do have some comments that need to be addressed.
- The authors claim (point 4, line 319) that it is an advantage to analyze splicing in a cell type relevant for the disease. I understand that they have used the MCF7 breast cancer cell line for other genes before, but it’s relevance for RAD51D seems less obvious because of its moderate effect on breast cancer risk. Also, cancer cell lines may well show specific defects in mRNA splicing (which may even be affected by culture conditions, PMID: 16488998). It is important to know if the minigene data in this manuscript accurately reflect splicing of endogenous RAD51D, ideally in normal ovarian or mammary epithelial cells and (because of the artificial nature of the minigene, with for instance truncated introns) also in MCF7 cells.
- The minigene transcripts are analyzed from both directions, which largely results in the same results. These data should be better explained: why are variants measured from a specific direction, are the differences between both directions indeed neglectable, why are the same wild-type results shown multiple times in figures 1C and 2B, what does ‘two assays x 3’ mean (also vs ‘six assays x 3’), what do the error margins represent, …
- Regarding the translation to an ACMG/AMP-like classification: it can be difficult to determine what may be deleterious and pathogenic, and the effort is much appreciated. For clarity, the footnote explanations of table 2 should be more concise (further explanation may be possible in the (SI) methods section). And it is not clear to me why the cut-off for effects on the wild-type transcript is set at 10%. Also because heterozygous mutation carriers still have a wild-type, functional RAD51D allele. The rationale should be explained, or it should explicitly be acknowledged that this is an arbitrary cut-off.
- The title of the manuscript somewhat misleadingly describes the analysis of effects on mRNA splicing as functional assays. By itself, this type of analysis does not measure effects on gene function. Although the text of the manuscript clarifies that functional analysis should be interpreted in the context of mRNA splicing, it should be acknowledged in the discussion that additional analyses are required to assess consequences for gene function.
Author Response
Response:
Thank you very much for the positive comments and the thorough scrutiny of the manuscript.
The number of lines where the corrections have been introduced corresponds to the revised version of the manuscript (Manuscript_RAD51D_Cancers_R1-Track_Changes.docx) in which all the changes can be tracked. All changes have been also highlighted in the final manuscript file.
We understand that the section 3.3 of ESE mapping and detection of ESE-variants is complex. For this reason, we have thoroughly modified the Supplementary Figure S1 (part B), which includes a workflow of the whole process, to clarify this point.
1.The authors claim (point 4, line 319) that it is an advantage to analyze splicing in a cell type relevant for the disease. I understand that they have used the MCF7 breast cancer cell line for other genes before, but it’s relevance for RAD51D seems less obvious because of its moderate effect on breast cancer risk. Also, cancer cell lines may well show specific defects in mRNA splicing (which may even be affected by culture conditions, PMID: 16488998). It is important to know if the minigene data in this manuscript accurately reflect splicing of endogenous RAD51D, ideally in normal ovarian or mammary epithelial cells and (because of the artificial nature of the minigene, with for instance truncated introns) also in MCF7 cells.
Response:
Acknowledge this comment.
It is true, as the reviewer suggests, that we have used MCF7 in part because it is a robust experimental system for minigene assays (Acedo et al 2015, Fraile-Bethencourt et al 2017, 2018, 2019a, 2019b). Yet, it is also true that MCF7 (breast cancer cell line) is relevant for RAD51D studies. According to the most recent analysis of cancer risk in carriers of RAD51D pathogenic variants (Yang et al, 2020), the RR for tubo-ovarian carcinoma in carriers RR=7.60, (95% CI=5.61 to 10.30) is much higher than the RR of breast cancer RR=1.83, 95% CI (1.24 to 2.72). Yet, the same study concludes that the absolute risk of developing breast cancer 20% (95% CI=14% to 28%) is higher than the absolute risk of developing ovarian cancer, 13% (95% CI=7% to 23%), highlighting the relevance of breast cancer disease in RAD51D carriers.
Further, in the present study we have analyzed RAD51D genetic variants in the framework of an association study in consecutive breast cancer cases and controls conducted by the breast cancer association consortium (BCAC) (Dorling et al, 2021).
We have added a new paragraph to the discussion section (Discussion, lines 336-351) to clarify this point.
“In relation with the latter, we have performed RAD51D minigene analyses in two breast cancer cell lines (MCF7 and MDA-MB-231), the latter derived from a triple-negative breast cancer (TNBC) patient. We assayed eight spliceogenic … … but not with ER-positive breast cancer risk [5,6]. Germ-line defects in all four genes are associated as well with increased ovarian cancer risk (for BARD1 the evidence is limited) [3,41-43].”
*Cancer cell lines
On the other hand, it is true, as the reviewer suggests, that cancer cell lines can show specific splicing defects. Most, if not all, cells used in minigene assays are tumor cell lines because they are easier to handle and results can be obtained in a shorter period of time.
In a previous report we checked the non-tumorigenic breast cell line MCF10A (Sanz et al 2010) and the result were very similar or identical. Moreover, this cell line grows very slow (even more after Lipofectamine transfection), so it is not feasible to perform a systematic study like the present one.
- * We have introduced three new sentences, Discussion, lines 357-363:
“On the other hand, minigene assays are commonly performed in tumor cells (MCF-7, HeLa, HepG2, HEK 293, etc.) because they are easy to handle and generate highly repro-ducible splicing patterns. However, it is known that the splicing machinery of cancer cells may be affected [45]. In this regard, in a previous study we ruled out the use of non-tumor breast cells (MCF-10A) since the splicing outcomes were very similar to those of tumor cells, and showed clear disadvantages, such as very slow growth or high cell lethality after transfection [46].”
*Splicing of endogenous RAD51D
-It is really difficult to confirm if the data of the manuscript or any other published paper reflect the splicing of endogenous RAD51D. Even working with patient RNA, there have been shown substantial differences between experiments and different laboratories: for instance, BRCA2 c.7976+5G>T (PMID: 29969168) or PALB2 c.3113+5G>C that showed three different results in three different papers (PMID: 17200671, PMID: 30890586, PMID: 32133419). We have to highlight that the RAD51D gene is linked to high levels of alternative splicing (Brandao et al 2019, so the amount of the full-length transcript may be linked to variations depending on experimental conditions (RNA isolation, RT-PCR, growth media, primer pairs, type of cell, thermocycling protocol, etc). Also, a comprehensive characterization of RAD51D alternative splicing profile in non-malignant breast and fimbriae tissues has discarded major tissue specific effects. Anyway, in general, in our hands the minigene assay mimics the splicing patterns of alternative splicing. As it is indicated in Results and the Figures, we have detected several alternative transcripts previously described by Brandao et al (2019), for example, we have also identified ∆(E3), ∆(E2_5), ∆(E3_5), ∆(E4_5), ∆(E3_7), but other minor peaks were observed that probably reflected other alternative transcripts. Moreover, the wt minigene have been assayed many times (much more than the mutant minigenes), and the standard deviations of the transcripts are small (less than 5.6%) indicating that the test is robust and reproducible. Our protocol has been thoroughly optimized in hundreds of tests. Furthermore, we have shown the high reproducibility of variant outcomes of minigene vs. patient RNA: Acedo et al 2015, Fraile-Bethencourt et al, 2017, 2018, 2019a, 2019b, Sanoguera-Miralles et al 2020. Thus, patient RNA outcomes of 38 variants were replicated in minigenes.
We can therefore conclude that the minigene strategy is an optimal tool for the initial characterization of splicing anomalies induced by germline variants.
- To clarify these points we have added/modified two paragraphs in Discussion, lines 352-363 and 366-379:
“Mounting epidemiological and molecular evidence indicates that high-grade serous ovarian cancer arises not from ovarian epithelial cells, but from cells in the fimbriae of the fallopian tubes [44]. Remarkably, a comprehensive characterization of RAD51D … … In this regard, in a previous study we ruled out the use of non-tumor breast cells (MCF-10A) since the splicing outcomes were very similar to those of tumor cells, and showed clear disadvantages,…”
“…some previous works from our group with BRCA2 or RAD51C minigenes [16,24,25,43–45], where patient RNA results of 38 variants were replicated in minigene assays. … … Therefore, minigenes appear as a valuable and sensitive tool to initially assess splicing outcomes, providing essential information to interpret spliceogenic variants.”
2. The minigene transcripts are analyzed from both directions, which largely results in the same results. These data should be better explained: why are variants measured from a specific direction, are the differences between both directions indeed neglectable, why are the same wild-type results shown multiple times in figures 1C and 2B, what does ‘two assays x 3’ mean (also vs ‘six assays x 3’), what do the error margins represent, …
Response:
- To increase PCR specificity of the cDNA, we used a primer located at one exon insert and another one at the vector exons V1 or V2. Thus, we used two primers pairs: [RAD51D exon 2-forward / Vector Exon V2-reverse] and [Vector exon V1-forward / RAD51D exon 9-reverse]. As indicated in Materials and Methods, exons 3 to 9 were analyzed with and RT-PCR forward primer in exon 2 and a reverse primer in V2 exon. Therefore, to check exon 2 variants, we used a different primer pair: forward primer in V1 exon and reverse primer in exon 9.
Thus, we can ensure that all the retrotranscribed/amplified cDNA is generated by the minigene RNA.
Materials and Methods, section 2.4. To clarify it, we have added the following sentence (lines 142-144):
“To increase specificity, cDNA was amplified with one primer located in one RAD51C exon of the insert and another one in a vector exon (V1 or V2).”
- Wild type results in Figures 1 and 2.
- The results of the wild type minigenes are shown several times to have the possibility to compare transcript sizes and observe their differences given that all electropherograms are aligned.
* What does ‘two assays x 3’ mean (also vs ‘six assays x 3’)
While all the variants have been tested in triplicate, the wt minigene have been assayed much more times, since, for example, any variant assay implies a wt minigene transfection in triplicate.
These data are not very informative, so we have deleted both.
* Error margins:
They are standard deviations that were calculated from three independent experiments of each variant (Included in Materials and Methods, last sentence of section 2.4. Functional Assays).
3- Regarding the translation to an ACMG/AMP-like classification: it can be difficult to determine what may be deleterious and pathogenic, and the effort is much appreciated. For clarity, the footnote explanations of table 2 should be more concise (further explanation may be possible in the (SI) methods section). And it is not clear to me why the cut-off for effects on the wild-type transcript is set at 10%. Also because heterozygous mutation carriers still have a wild-type, functional RAD51D allele. The rationale should be explained, or it should explicitly be acknowledged that this is an arbitrary cut-off.
Response:
Footnote explanation is now more concise. We have moved some explanation to:
- footnote of Table 1
“According to ref. 33, c.738+1G>A causes only a 37-nucleotide intron retention ▼(E8q37). Yet, some methodological issues (location of PCR primers not reported, RT-PCR products analyzed only by low sensitivity agarose electrophoresis, no agarose band isolation before Sanger sequencing, and no allele-specific information) preclude, in our opinion, direct comparison with our minigene result”
- and Discussion section (lines 477-480 and 497-503)
“we do not intend to provide here (see Table 3) a definitive clinical classification of genetic variants, but rather to highlight some limitations of the current ACMG/AMP framework to interpret complex spliceogenic outcomes (as those observed for several RAD51D variants, see Tables 1 and 2)”
“Current ACMG/AMP guidelines [29] have not been developed to identify “intermedi-ate risk variants”. Yet, we think is worth considering this possibility for certain variants expressing variable proportion of (likely) functional and (likely) non-functional mRNAs (see Table 3 and Supplementary Methods). It will be extremely challenging to evaluate risk for individual RAD51D variants that are exceedingly rare in the population, but it should be possible in principle to design burden analyses for variants displaying similar splicing effects.”
10% Cut-off
We have to take into account that: i) the wild type minigene produces fl-transcript and physiological alternative transcripts; ii) mainly, all but one exon 3 variants induce partial splicing effects (we mean that the FL-transcript is detected). Therefore, we had to define a cut-off to classify variants under the splicing impact viewpoint.
Given that the standard deviation of the full-length transcript proportion in the wild type minigene is ±5.6 (equivalent to a 7.7% increase/reduction of the fl-transcript), we were conservative and established an arbitrary 10% reduction as the cut-off to consider that a variant affects normal splicing. This 10%-cutoff is not related with the putative pathogenicity or loss-of function of a particular variant.
- A reference to the <10% cut-off has been removed from the results section on canonical splice-site variants, as it is not relevant. All 11 splice-site variants show a reduction of the FL-transcripts contribution >>10%
- The 10% cut-off has been exclusively used to identify spliceogenic variants targeting exon 3 ESEs. In the revised version, the cut-off definition has been improved (result section, lines 244-245):
“In this study, variants with a FL-transcript proportion ranging from 65.8%-73.1% (<10% reduction of the FL-transcript in the wt minigene) were not considered spliceogenic (see Discussion).”
- Now, the rationale to include an arbitrary 10% cut-off for effects on the wild-type transcript has been explicitly acknowledge and explained in the discussion section (lines 409-418):
Of note, in a region with high levels of naturally occurring alternative splicing and most variants with limited impacts, it is controversial the definition of a spliceogenic variant. In this study, we have defined an arbitrary 10%-reduction cut-off for the contribution of the FL-transcript to the overall expression (i.e. variants with 65.8% to 73.1% of the FL-transcript are not considered spliceogenic). Based on this cut-off criteria, 30 variants (71.4%; Table 2) impaired splicing, thus supporting our SRE-screening strategy. All spliceogenic SRE-variants but one (c.202G>A), produced non-negligible levels of FL-transcripts (65.1% to 27.4%), with variants c.216C>G, c.215A>C, c.200_218del, and c.187G>C producing 36.4%, 28.7%, 28.7%, and 27.4%, respectively. c.202G>A was the only tested SRE-variant that did not generate FL-transcripts.
4- The title of the manuscript somewhat misleadingly describes the analysis of effects on mRNA splicing as functional assays. By itself, this type of analysis does not measure effects on gene function. Although the text of the manuscript clarifies that functional analysis should be interpreted in the context of mRNA splicing, it should be acknowledged in the discussion that additional analyses are required to assess consequences for gene function.
Response:
To avoid misunderstanding of “functional”, we have modified the title to
“RAD51D aberrant splicing in breast cancer: identification of splicing regulatory elements and minigene-based evaluation of 53 DNA variants”
Further, now we have added to the discussion section (lines 494-496) the following sentence “Indeed, for this and other RAD51D variants predicted to introduce missense changes and/or in-frame alterations, additional analyses are required to assess consequences for gene function”.
Round 2
Reviewer 2 Report
The authors have addressed my concerns and sufficiently explained why additional analysis in normal ovarian or mammary epithelial cells is not feasible and likely not required. I would be happy to see this work published and only have a few minor remarks:
- Please correct the new statement in lines 409-411 "..., it is controversial the definition of a spliceogenic variant." to something like "..., the definition of a spliceogenic variant is controversial." Also for other parts of the text some language editing is needed.
- The comments on relevance of this study in relation to breast and ovarian cancer risk and the lack of evidence for major tissue specific differences in splicing in lines 336-363 could be more concise to improve the flow and the focus of the discussion. For instance, mentioning that the absolute risk to develop breast cancer is higher than that for ovarian cancer does not seem very relevant in the context of this study. Some brief comments on risk associations could already be made in the introduction and the validity of the model system should not need a lengthy explanation. I do like the additional references.
Author Response
We acknowledge the positive comments of the reviewer.
We have added a new sentence in the Introduction. We have corrected the statement of lines 409-411.
We have modified and summarized several paragraphs of the Discussion according to her/his suggestion.
All changes are yellow-highlighted.